# Resilience, Satisfaction with Life and Anxiety in the Israeli Population after Mass Vaccination for COVID-19

**DOI:** 10.3390/healthcare12020255

**Published:** 2024-01-19

**Authors:** Miriam Sarid, Rony Tutian, Maya Kalman-Halevi, Sharon Gilat-Yihyie, Adi Sarid

**Affiliations:** 1Department of Education, Western Galilee College, Acre 2412101, Israel; ronyt@wgalil.ac.il (R.T.); mayah@wgalil.ac.il (M.K.-H.); sharong@wgalil.ac.il (S.G.-Y.); 2Sarid Institute, Haifa 2626047, Israel; adi@sarid-ins.co.il

**Keywords:** anxiety, COVID-19 vaccine, employment status, resilience, socio-economic status

## Abstract

Objectives: The current study aimed to examine the relationships between resilience and personal characteristics such as socio-economic status, employment, satisfaction with life, and anxiety, during the period of returning to routine life after mass vaccination for COVID-19. Method: 993 Israeli participants, 52% female and 48% male, with a mean age of 40 years (18 to 89 years old) responded to an online questionnaire during March 2021. We hypothesized that (1) unemployed individuals and those with a low SES would have lower resilience, lower satisfaction with life and higher anxiety, (2) individuals who were ill with COVID-19 would have lower resilience and satisfaction with life and a higher level of anxiety, and (3) a higher resilience would be related to a lower level of anxiety. Results: The findings showed that unemployed individuals had lower levels of resilience and satisfaction with life and higher levels of anxiety than employed individuals. Specifically, those who experience a large gap between their socio-economic and employment statuses are at a greater risk than others. In addition, differences were found between people who had experienced COVID-19 illness and those who had not, but only with their satisfaction with life. People who had been ill were more satisfied than those who had not been ill. Eventually, as expected, a higher resilience was related to a lower level of anxiety, specifically at the lower levels of resilience. Conclusions: The findings of this study can provide additional perspectives on the day after a crisis (COVID-19) and the need for the development of intervention programs to strengthen the resilience of individuals who experience a gap between their SES and employment statuses when returning to their routine life after a crisis. The study also shed light on the unique correlation between anxiety and resilience, implying that following a crisis, high-resilience individuals face their anxiety better than low-resilience individuals.

## 1. Introduction

At the beginning of the COVID-19 pandemic, the assumption in Israel was that the public was mostly concerned with health issues rather than political or economic consequences [1]. A serious illness or injury might be a life stressor that affects the psychological well-being of an individual [2,3]. After a year of the COVID-19 pandemic, at the end of the third lockdown, a high percentage of the population in Israel had been vaccinated, and the infection rate had dropped [4], and Israelis were resuming a more regular everyday life. However, soon after that, during the summer of 2021, the unemployment rate in Israel increased to its peak of 5.5% [4]. Studies have shown that economic hardships such as unemployment may affect people’s mental state [5], resilience, and distress symptoms [6]. The findings of these studies highlight the importance of understanding the effect of the economic and health factors that accompanied the COVID-19 pandemic on the human psychological state (i.e., resilience, satisfaction with life, and anxiety). Resilience is defined as a stable trajectory of healthy functioning after a highly adverse event [7]. Another measure that has been linked to resilience is satisfaction with life. Satisfaction with life is a construct that reflects well-being and is characterized by an individual’s positive evaluation of their life experiences [8]. Anxiety is another emotional state that represents the distress associated with uncertainty and existential concerns [9].

Most of the research on the emotional implications of COVID-19 was conducted during the crisis, but research on the recovery process, when the health threat of the pandemic had decreased and people had to return to their routine lives, is scarce. The period just after the mass vaccination for COVID-19 may be seen as a case study of returning to healthy functioning after a highly adverse event. Therefore, the current study examined the relationship between resilience, satisfaction with life, and anxiety with economic statuses (SES and employment) and being infected by COVID-19 during the pandemic outbreak. Three hypotheses were examined and are as follows:(1)It was hypothesized that unemployed individuals and those with a low SES would have lower resilience, lower satisfaction with life, and higher anxiety.(2)It was hypothesized that individuals who were ill with COVID-19 would have lower resilience and satisfaction with life and a higher level of anxiety.(3)It was hypothesized that higher resilience would be related to a lower level of anxiety.

## 2. Theoretical Review

Resilience, as defined by the American Psychological Association, is the ability to recover and adjust effectively in the aftermath of challenging experiences, as well as in the presence of trauma, tragedy, dangers, or substantial stressors [10]. It refers to the capacity of adults in typical situations who experience a solitary and potentially a very disruptive event, like the loss of a close family member or a life-threatening situation, to sustain relatively stable and healthy levels of psychological and physical well-being [7].

The significance of the concept of resilience stems from its direct correlation with the capacity to promptly and effectively react to unforeseen events [11], hence facilitating adjustments to evolving circumstances [12]. The COVID-19 epidemic can be seen as an example of a worldwide crisis. The mentioned factors, such as uncertainty, ambiguity, the loss of control, social isolation, and concerns about own health and the health of loved ones, are recognized to induce stress and emotional distress [13]. These can manifest as internalizing symptoms like anxiety, despair, and anger [14,15]. Amidst the COVID-19 pandemic, individuals must effectively handle persistent stimuli and strive to minimize psychological suffering [16].

In addition, a serious illness or injury and the death or illness of close acquaintances are other life stressors that may also decrease resilience [2], as examined in post-traumatic symptoms related to the emergency itself [17]. High-resilience individuals have been found to successfully overcome trauma and crisis and return to their routines [18], while low-resilience individuals are more vulnerable to mental distress and to the development of psychological sequelae [19].

Subjective well-being, a personal assessment of one’s own quality of life and health, has also been linked to resilience. This construct of well-being is characterized by an individual’s positive evaluation of their life experiences [20] and reflected by indices of satisfaction with life (SWL). During the COVID-19 pandemic, there was a significant and inverse relationship between subjective well-being and the experience of danger and symptoms of distress [21]. Individual resilience was found to be related to having a higher sense of emotional well-being [8,22]. The research shows that positive emotions can foster resilience [23,24] because they facilitate flexible thinking [25] and encourage both adaptive coping [26] and the maintenance of social relationships with family, friends, or significant others [27]. Well-being was also found to predict anxiety [6].

Demographic factors that may reduce well-being are unemployment and chronic illness [17].

An extensive study [28] conducted after the September 11 attack revealed that having a consistent income was a noteworthy indicator of resilience, even when accounting for other socio-economic and demographic factors. Individuals who had a substantial decrease in their income were just half as likely to exhibit resilience compared to participants who did not experience any income loss [28].

Income loss is seen as a significant source of stress in life, and being exposed to it might hurt an individual’s ability to cope emotionally in the future. When people are physically safe and their basic needs are looked after, this is thought to increase a sense of control and emotional security [29]. Economic hardships resulting from the COVID-19 pandemic have been significantly and positively correlated with the increased levels of distress, as identified in research [6]. In contrast, the research outlined by [2] indicates that participants who remained employed during the pandemic exhibited significantly reduced symptoms of stress, anxiety, and depression.

In the context of COVID-19 pandemic, anxiety emerges as a predominant emotional response [19]. This aligns with the Cognitive Appraisal Theory, which posits that emotions are a product of the interplay between individuals and their surroundings [29]. Anxiety fundamentally represents the distress associated with uncertainty, holding existential weight that extends beyond immediate threats to encompass concerns about one’s ability to thrive [30].

Anxiety is often explained in terms of its adaptive significance, particularly when it is mild. The accompanying physiological and psychological arousals can help the organism stay alert and eventually recognize danger, leading to appropriate coping actions. Therefore, anxiety can be advantageous in terms of enhancing alertness. However, if it becomes too intense, it may have the opposite effect, narrowing attention to cues that are irrelevant for appropriate solutions [31]. Therefore, anxiety is not a sign of weakness; it is an understandable feeling in response to unpredictable situations.

Previous research has revealed that during pandemics, people feel anxious about getting sick and dying [6,9]. Many individuals wondered whether they had already been impacted by the COVID-19 virus, while not exhibiting the characteristic symptoms associated with it [32]. A meta-analysis, which included 59 studies (19 conducted before the pandemic, 37 during the pandemic, and 3 that were conducted both before and during the pandemic), compared negative emotions such as anxiety, depression, and stress from before the pandemic to those during the pandemic and revealed that globally, these negative emotions were elevated during the pandemic compared to those before the pandemic [33].

In an analysis conducted within Israel [34], adults exhibited moderate-to-high levels of general stress and concern during the pandemic, aligning with the pervasive uncertainty and stringent restrictions that implied the severity of the crisis. Despite these factors, reported anxiety levels remained low, a phenomenon which may be ascribed to the resilience inherent to the Israeli society, attributed to decades of exposure to wars and ongoing political unrest, potentially creating a populace adept at managing stress. Supporting this, research has demonstrated a significant inverse relationship between individual resilience and anxiety [6].

Beyond its impact on daily life, the hallmark of the COVID-19 pandemic has been uncertainty. The combination of uncertainty and the rapid spread of illness created stress and even blocked the possible sources of protection and emotional resilience such as in-person contact with family and friends [35]. A meta-analysis that included 44 published studies across 20 countries revealed that the prevalence of low resilience in the general population during the COVID-19 pandemic was 35%. This indicates that nearly 4 out of every 10 individuals in the general population had low resilience [19]. The COVID-19 pandemic has upended normal existence for much of the world’s population, including that of Israel.

### The Current Study

Many studies investigated resilience, satisfaction with life, and anxiety during COVID-19. The Israeli population underwent a quick and extensive vaccination program at the end of the third lockdown, which enabled a decline in infection rates and a process of recovery from the pandemic-related situation. As far as we know, no research exists about the resilience of the population while returning to their routine lives after COVID-19.

The current study aimed to examine the emotional resilience, satisfaction with life, and anxiety of the population in Israel and their correlates (i.e., socio-economic and employment statuses and being infected with COVID-19) at a point in time when the danger of illness had been reduced by the vaccine, and it seemed that everyday life was returning to a more normal routine, albeit with some restrictions that did not exist before the start of the pandemic.

Therefore, our hypotheses were as follows:

**H1:** *As socio-economic security is related to a greater ability to recover from crises, we hypothesized that unemployed individuals and those with a low SES would have lower resilience and satisfaction with life and higher anxiety*.

**H2:** *As being infected with COVID-19 can disrupt emotional recovery from the crisis, we hypothesized that individuals who were ill with COVID-19 would have lower resilience and satisfaction with life and a higher level of anxiety*.

**H3:** *It was hypothesized that higher resilience would be related to a lower level of anxiety*.

## 3. Method

### 3.1. Participants and Sampling

The total sample included 993 native Hebrew-speaking adults from Israel between the ages of 18 and 89 (mean = 40 years, *SD* = 13.96). There were 474 male participants (48% of the total) and 518 female participants (52%). Regarding marital status, 615 (62%) of participants were married, and 378 (38%) were single (see Table 1). The sample was recruited from an online panel in Israel representing the adult population in Israel, with special emphasis on obtaining a representative sample according to gender, age, socio-economic status, and sampling from varying cities and residential areas in Israel. The data were collected during one week in the middle of March 2021.

### 3.2. Instruments

#### 3.2.1. Resilience (CD-RISC)

The questionnaire is a shortened version of 10 items from the original version of the 25-item resilience scale [36] and includes one factor of resilience with a high internal consistency (alpha = 0.92 in the current research). All items are scored on a Likert scale of 1 (lowest agreement) to 5 (highest agreement) and averaged into a mean score.

#### 3.2.2. Generalized Anxiety Disorder Screener (GAD-7)

The Generalized Anxiety Disorder (GAD-7) [37] scale is a 7-item, self-report anxiety questionnaire designed to evaluate the respondent’s anxiety status. The items inquire about the degree to which the respondents would describe themselves as being “nervous, anxious, or on edge”. The items were rated on a scale from 1 (lowest agreement) to 4 (highest agreement). All the items were averaged to a mean score. The internal reliability of the current research was 0.95.

##### Satisfaction with Life Scale (SWLS)

The SWLS is a 5-item index [38] designed to assess the overall cognitive assessment regarding satisfaction with life (well-being), which signifies positive life outcomes [38]. The scale includes items such as “So far, I have achieved the important things I want in life”. The items were rated on a scale of agreement from 1 (strongly disagree) to 7 (strongly agree) and were averaged to a mean score. The reliability of the translated questionnaire was 0.82. In the present study, the internal reliability for the entire questionnaire was 0.90.

### 3.3. Demographic Variables

The demographic variables questionnaire included general characteristics such as gender and age, as well as questions regarding socio-economic status (SES): (1) participants’ self-ranking of their income as compared to their family average net income in Israel, representing two ranks below average, one rank of average, and two ranks above average (very much below average, below average, average, above average, and highly above average) and (2) occupational status before the pandemic and at the end of the third outbreak during the vaccination of the Israeli population.

### 3.4. Statistical Analysis

To examine the distribution of the sample’s demographic characteristics, descriptive statistics were obtained. The first hypothesis about the impact of SES and employment statuses on resilience, anxiety, and satisfaction with life was investigated with two-way Analysis of Variance (ANOVA). An independent samples *t*-test was utilized to compare resilience, anxiety, and satisfaction with life between individuals who were infected and individuals who were not infected with COVID-19 during the pandemic. The third hypothesis that aimed to examine the correlation between resilience and anxiety was examined by Pearson correlation for the whole sample and for subgroups of high, medium, and low levels of resilience. The differences in the correlations between those groups were tested with a Fisher test comparing the statistical significance of the difference between Pearson correlations.

## 4. Results

H1, the first hypothesis, posited that people with a low SES and the unemployed will have lower resilience and satisfaction with life and higher anxiety. Table 2 presents the results of two-way ANOVA that examined the hypothesis.

The findings show that employed participants had higher scores on all three measures (e.g., anxiety, resilience, and satisfaction with life) compared to unemployed participants (see Table 2). In addition, there were differences in satisfaction with life according to the socio-economic status, showing that low-income participants’ satisfaction with life was lower than the satisfaction with life of those with average or higher than average SES, *F*(1,988) = 30.8, *p* < 0.001.

In addition, the differences between employment status groups were significant in resilience, anxiety, and satisfaction with life. Participants who were not employed exhibited higher levels of anxiety than employed participants, *F*(1,988) = 31.3, *p* < 0.001, and lower resilience scores, *F*(1,988) = 16.5, *p* < 0.001. Their satisfaction with life was lower than the satisfaction with life of employed participants, *F*(1,988) = 30.8, *p* < 0.001.

An interaction effect of SES and employment, *F*(1,988) = 5.19, *p* = 0.02, indicated that individuals with average and higher socio-economic statuses who were not employed exhibited lower resilience than participants in the same SES who were employed, while employed and unemployed participants with a lower socio-economic status did not differ in their resilience level.

In H2, it was hypothesized that participants who had been ill with COVID-19 would have lower resilience and satisfaction with life and higher levels of anxiety. Differences were found between individuals who had been ill and those who had not been ill, but only in SWL (see Table 2). Individuals who had been ill and recovered from COVID-19 were more satisfied with their lives than others who had not been ill, *T*(991) = 3.30, *p* < 0.001.

In H3, it was hypothesized that higher resilience would be related to lower levels of anxiety. Pearson correlations for the whole sample confirmed this hypothesis and revealed that higher resilience is related to a lower level of anxiety.

The relationships between resilience and anxiety were also examined among resilient and less resilient participants (see Table 3). We classified the participants into three groups based on their resilience scores. The first group included the lowest 25 percent of participants in resilience scores (cut-off score: 3.3, *n* = 281), and the highest group included the highest 25 percent of participants (above cut-off score: 4.0, *n* = 256) in their resilience score. All other participants were half of the sample (*n* = 456), who earned medium scores in resilience. A negative correlation between anxiety and resilience scores was found in the lowest resilience group, *r* = −0.35, *p* < 0.001. In the group of medium resilience scores, a weaker correlation was found between resilience and anxiety, *r* = −0.15, *p* < 0.001. No correlation was found between resilience and anxiety in the most resilient group, *r* = 0.06, *p* = 0.18. Examining the differences between correlations by Fisher test indicated that the correlation between the least resilient individuals was found to be different when compared with the highly resilient (Z= −4.90, *p* < 0.001) and medium resilient groups (Z= −2.81, *p* = 0.002).

## 5. Discussion

The current study examined resilience, satisfaction with life, and anxiety in the Israeli population after mass COVID-19 vaccination. The results of the study indicated that unemployment status was related to lower resilience, higher anxiety, and lower satisfaction with life. These results exhibit that the pandemic created not only a health crisis but also emotional issues related to coping with the secondary impacts of the crisis, i.e., economic hardships and financial insecurity created by unemployment. Individuals who faced a gap between their prior high SES and their unemployment status during the crisis were less resilient than others, a finding that may point to a greater difficulty in recovering when one has a high starting point.

The first hypothesis addressed the differences between socio-economic status groups and employment status in resilience, satisfaction with life, and anxiety. The resilience and satisfaction with life of unemployed individuals were lower than those variables among employed individuals, and their anxiety level was higher. It is not a surprising finding, as employment represents one of the possible significant perceived psychological losses [39]. Additionally, Lipskaya-Velikovsky [27] discovered that individuals who were working during the pandemic reported much lower levels of stress, anxiety, and depression. Some of the participants in our study either lost their jobs or perhaps were anxious about finding a job in this time of financial crisis. It might suggest that the pandemic has both financial security and health costs. Furthermore, as these participants felt more anxious and less resilient, it may indicate that, besides the health threat, the pandemic may have emotional costs that are related to the lockdown and the closure of many workplaces.

The interesting finding in our study was the interactive effect of socio-economic status and employment on resilience. The resilience of unemployed individuals with average and higher socio-economic statuses decreased more than the resilience of those who are unemployed with a lower socio-economic status. It may be that the combination of unemployment and high socio-economic status creates a larger gap between the present circumstances and the quality of life that the individual is used to. Individuals from low SES were used to living on a reasonable budget, and thus, this period of unemployment may have created a smaller gap between their habits and actual status and therefore affected their resilience to a lesser extent. Moreover, Matiz et al. [40] suggest that although losing a job is unpleasant, if people can prepare themselves for the situation and think they can cope with it, this event is less likely to cause stress, anxiety, and depression. It may be that unemployed individuals who are used to a high level of income at regular times experience the greatest deterioration of their quality of life and an elevation of anxiety. The emergence of these findings upon the resumption of normal activities suggests the potential benefit of developing intervention programs such as mindfulness-based training [40] for at-risk individuals following the conclusion of a crisis.

The second hypothesis addressed the differences between individuals who were infected with COVID-19 and individuals who were not infected. In contrast to our hypothesis and previous study [23], it was found that participants who had been ill with COVID-19 were more satisfied with their lives than those who had not been ill. These findings are supported by recent research that examined the relationship between resilience and satisfaction with life [41]. According to the author’s conclusions, it may be that individuals who have not been infected may experience higher levels of fear and anxiety related to the virus, which could contribute to lower satisfaction with life.

It may be assumed that surviving the illness may be a source of successful coping in the COVID-19 pandemic. It may allow the individual to see the future in a more optimistic way and provide a sense of “been there, done that”, enabling them to view COVID-19 as another obstacle they overcame. These individuals feel that they are not endangered anymore. It may be that having been ill provides them a sense of victory over the pandemic and the feeling of being naturally protected with antibodies against the virus as if they were vaccinated.

The third hypothesis posed whether individuals with higher resilience will express lower anxiety. As expected, we found a negative correlation between resilience and anxiety, showing that less resilient individuals are more anxious. These results are also supported by recent studies conducted during COVID-19 among the general population in Israel [18,34].

When classifying participants into three groups of resilience (low, middle, and high), a negative correlation was found mainly in the low resilience and middle resilience participants but not in the high resilience group. It may indicate that the high resilience group reflects more adaptive and integrative emotion regulation (allowing the exploration and experience of emotions) and not suppressive regulation (avoiding or minimizing the experience of negative emotions) [42,43]. The self-determination theory perceives the negative emotions that individuals face as a source of important information and a deeper understanding of the situation regarding the individual’s condition and their goals [42,43]. According to Lazarus [44], each emotion arises from a different plot or story about relationships between a person and the environment. Emotions that are perceived as negative (e.g., anxiety) are not less important than positive emotions, as through these emotions, individuals can learn a lot about themselves, their situation, and ways to adapt to it [44]. This assumption regarding the relationship between the level of resilience and the types of emotion regulation should be further investigated.

## 6. Implications

The COVID-19 pandemic presented psychological and economic difficulties for Israelis in addition to health concerns. The findings of the present study highlight that the people may be at risk of expressing low resilience or low satisfaction with life on a “day after” a crisis. As a high-risk population, unemployed high-SES individuals could benefit from preventive programs. Those individuals may have greater economic commitments and quality of life, and therefore for them, the decrease in economic efficacy may be more profound. Governmental intervention programs may be crucial in supporting these individuals and helping them return to the employment cycle.

## 7. Limitations

The present research was undertaken during an unprecedented time in the pandemic of COVID-19, which presented a unique context for the study. Nevertheless, the lack of pre-existing data on resilience, satisfaction with life, and anxiety within the Israeli populace was a limitation. Moreover, due to the design of the current study, which is not longitudinal, causal conclusions cannot be drawn. Therefore, future research should incorporate a longitudinal methodology to thoroughly investigate these variables.

## 8. Conclusions

The main conclusion of the current study is that the response to a crisis may differ according to unique characteristics of the population facing the crisis. Beyond the direct health ramifications and the accompanying emotional aftermath, individuals were compelled to grapple with economic consequences linked to these emotional states. The degree of personal resilience and the ability to manage a crisis with extensive economic impacts are significantly affected by the degree of decline in their quality of life due to the crisis. This observation highlights the importance of alterations in the initial life quality as a resilience factor and the critical importance of re-entering the employment cycle.

The unique correlation between resilience and anxiety found in the current research implies that individuals with high resilience may allow themselves to experience anxiety, explore it, and learn from their emotional state about themselves and their situation. It may be suggested that resilience is reflected in the ability to face negative emotions rather than avoid or minimize them. This relationship should be further examined.

## Figures and Tables

**Table 1 healthcare-12-00255-t001:** Demographic characteristics of participants.

Demographic Characteristics		N (%)
Gender	Male	474 (48%)
	Female	519 (52%)
Level of income	Below average	527 (53%)
	Average	284 (29%)
	Above average	181 (18%)
Level of education	12 or fewer years	290 (29%)
	13–14 years	247 (25%)
	Academic degree	456 (46%)
Marital status	Single	378 (38%)
	Married	615 (62%)
Occupational status	Unemployed	179 (18%)
	Employed	814 (82%)
Being infected by COVID-19	Yes	887 (90%)
	No	102 (10%)

**Table 2 healthcare-12-00255-t002:** Resilience, satisfaction with life, and anxiety according to socio-economic and employment statuses.

SES	Employment Status	Anxiety		Resilience		Satisfaction with Life	
		Mean	*SD*	Mean	*SD*	Mean	*SD*
Low SES	Unemployed	7.48	6.85	3.60	0.74	4.69	1.50
	Employed	5.48	5.59	3.70	0.65	5.13	1.21
	Total	5.91	5.93	3.68	0.67	5.04	1.29
Average+	Unemployed	7.65	6.22	3.46	0.70	4.78	1.32
	Employed	4.36	4.98	3.81	0.61	5.46	1.04
	Total	4.83	5.30	3.76	0.64	5.37	1.11
Total	Unemployed	7.54	6.61	3.55	0.73	4.72	1.43
	Employed	4.93	5.33	3.75	0.63	5.29	1.14
	Total	5.40	5.67	3.72	0.65	5.19	1.22
Employment *F*(1,988)		31.3 ***		16.6 ***		30.8 ***	
SES *F*(1,988)		0.98		0.16		4.34 *	
Interaction Employment × SES *F*(1,988)		1.85		5.19 *		1.39	
Being infected with COVID-19	No	17.65	7.30	3.71	0.66	5.16	1.22
	Yes	17.35	7.55	3.78	0.64	5.54	1.07
*t*		0.37		1.06		3.30 ***	

* *p* < 0.05, *** *p* < 0.001. *SD* represents standard deviation. The table presents two-way ANOVA and *t*-test.

**Table 3 healthcare-12-00255-t003:** Pearson correlation coefficients between variables of study.

	Anxiety	Resilience	Mean (*SD*)
Satisfaction with life	−0.37 ***	0.54 ***	5.19 (1.22)
Anxiety		−0.32 ***	5.40 (5.66)
Resilience			3.72 (0.66)

*** *p* < 0.001, and *SD* represents standard deviation.

## Data Availability

The data presented in this study are not publicly available due to privacy restrictions.

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
