# Peer review of "Resilience, Satisfaction with Life and Anxiety in the Israeli Population after Mass Vaccination for COVID-19"

_healthcare, 2024, doi:10.3390/healthcare12020255_

Round 1
Reviewer 1 Report
Comments and Suggestions for Authors
Author Response
We are grateful for the opportunity to revise our manuscript. The detailed and constructive feedback from the reviewe has been extremely beneficial in improving our work. Below, you will find our thorough response to each of their comments.
Reviewer 1.
- Abstract – please correct the sentence of conclusion. "the findings of this study…" The sentence was corrected, please see abstract.
- Suggests to specify the risk in the conclusions
Thank you for the suggestion to focus on the risk. We revised and specified the population at risk.
- Participants and sampling - please indicate the minimum and maximum of the scales.
We added the minimum and maximum scores of the scales in the method – instruments section.
- Reference for the table 1 in the text; suggest to add the percent values next to the n values.
Thank you for noticing this, we added reference to table 1 in the section of participants and sampling, and moved the percent values next to the corresponding n.
- Please provide in this part statistical methods used in the analysis.
We thank the reviewer for noticing this missing paragraph. We added a section of statistical analysis before the results section.
- Not found reference for table 2.
We added a reference for table 2 below table 2.
- Please indicate under table 2 what statistical test was used and what means the SD. We added in a note below table 2 the type of the statistical test used and the meaning of SD.
- In raw 255 is p=18 is correct.
We added a period that was omitted by mistake, before the number 18.
- I have not found reference for table 3 in the text.
A reference for table 3 was added in the paragraph before table 3.

Reviewer 2 Report
Comments and Suggestions for Authors
The paper has a great actuality; however it needs clarification in several aspects.
1) Title: please add the information about the sample: among whom was the study conducted?
2) Abstract: Likewise, we need to know the place where the data were collected, "jewish" is not enough, lacking: age range, sex ratio, place and time of data collection etc.
Aim the study is not clear: since it is not a longitudinal study, we know nothing about the psychological status of the people pre-vaccination. In this case, the post-vaccination term is not significant. E.g., "People who were ill were more satisfied than those who were not ill." Is this referred to the post-covid vaccination period? " Conclusions: The findings of this study can provide an additional perspective on the day after... " - this is a new information, timing may be relevant here.
3) Lines 29-31: "The process of recovering from difficult experiences
and adapting well in the face of adversity or significant sources of stress is termed “resilience” [5]." This reference does not include a definition of reference and on the other hand, this is NOT a valid definition of resilience.
Hypotheses: again, it is not clear why post-covid vaccination is a relevant timing and how it is related to the hypotheses... it may be the lockdown period and the returing might be related to this period???
Lines 43-44: "The COVID-19 pandemic declared by the World Health Organization in March 2020 [5] has led to significant morbidity and mortality globally." This phrasing is awkward and meaningless.
In Introduction the authors did not write about the mnain topcic, post-covid vaccination and its role in resilience and other psychological factors.
Line 330: "The current research results suggest that the “day after”
the pandemic after the vaccination to COVID-19... " I cannot imagine what the authors want to say here...
Overall, I suggest that the authors should rethink what they want to present by this study.
Ple
Comments on the Quality of English LanguageIt needs clarification in some places.
Author Response
Dear Reviewer,
We are grateful for the opportunity to revise our manuscript. The detailed and constructive feedback from the review has been extremely beneficial in improving our work. Below, you will find our thorough response to each of their comments.
- Please add the information about the sample among whom was the study conducted.
Thank you for this comment. We added a note in the section of participants and sampling regarding sampling from the adult population in Israel.
- Abstract: Likewise, we need to know the place where the data were collected, "jewish" is not enough, lacking: age range, sex ratio, place and time of data collection etc.
We added the demographic characteristics of the sample in the abstract.
- Aim the study is not clear: since it is not a longitudinal study, we know nothing about the psychological status of the people pre-vaccination. In this case, the post-vaccination term is not significant. E.g., "People who were ill were more satisfied than those who were not ill." Is this referred to the post-covid vaccination period? "
The comparison referred to the difference between those two populations who experienced or did not experience being infected, regarding their satisfaction with life, retrospectively. The comparison aimed to examined whether being infected had an impact on their satisfaction with life. We addresed this limitation in the limitation section.
- Conclusions: The findings of this study can provide an additional perspective on the day after... " - this is a new information, timing may be relevant here.
We thank the reviewer for bringing to our attention the need to clarify this conclusion. We added a clarification in the abstract
- Lines 29-31: "The process of recovering from difficult experiences and adapting well in the face of adversity or significant sources of stress is termed “resilience” [5]." This reference does not include a definition of reference and on the other hand, this is NOT a valid definition of resilience.
The reference was replaced with another one and the definition was specified according to the new reference.
- Hypotheses: again, it is not clear why post-covid vaccination is a relevant timing and how it is related to the hypotheses... it may be the lockdown period and the returning might be related to this period???
Thank you for this notification. We refined the clarification why a study during this period is valuable.
- Lines 43-44: "The COVID-19 pandemic declared by the World Health Organization in March 2020 [5] has led to significant morbidity and mortality globally." This phrasing is awkward and meaningless.
The sentence and the paragraph were rephrased.
- In the Introduction the authors did not write about the main topic, post-covid vaccination and its role in resilience and other psychological factors.
We refined the need to investigate resilience at the post-covid era at the introduction.
- Line 330: "The current research results suggest that the “day after”
the pandemic after the vaccination to COVID-19... " I cannot imagine what the authors want to say here...
In order to clarify the sentence, we added the term COVID-19.
- Overall, I suggest that the authors should rethink what they want to present by this study.
We hope that the rephrasing of the introduction as well as the corrections of the rest of the comments made the manuscript clearer.

Reviewer 3 Report
Comments and Suggestions for Authors
In the manuscript, the authors reported a study that examined the association of resilience with characteristics of the sample. The findings are potentially important and have practical implications. However, there are several issues that need to be addressed.
1. It is necessary to state the hypotheses in the opening paragraph.
2. In the second paragraph on p.4, the aims of the study include investigating the relationship demographic characteristics and resilience. Those factors should be covariate, rather than variables of interest.
3. In the hypotheses stated on p.4, there is no hypothesis regarding well-being. Please clarify the inconsistency.
4. The section "Method" should be added to include details of the study.
5. In Table 1, the number of "Employed" is not recognizable.
6. Limitations and conclusion are missing.
Author Response
Dear Reviewer,
We are grateful for the opportunity to revise our manuscript. The detailed and constructive feedback from the review has been extremely beneficial in improving our work. Below, you will find our thorough response to each of their comments.
1. It is necessary to state the hypotheses in the opening paragraph.
We added the hypotheses in the opening paragraph
2. In the second paragraph on p.4, the aims of the study include investigating the relationship demographic characteristics and resilience. Those factors should be covariate, rather than variables of interest.
We rephrased the theoretical review and removed the description of demographic variables that were not investigated in this study.
3. In the hypotheses stated on p.4, there is no hypothesis regarding well-being. Please clarify the inconsistency.
We added satisfaction with life to the hypotheses.
4. The section "Method" should be added to include details of the study.
The title of the method section was added
5. In Table 1, the number of "Employed" is not recognizable.
The table was corrected.
6. Limitations and conclusion are missing.
We added a limitations and conclusion section.

Reviewer 4 Report
Comments and Suggestions for Authors
Please see attachment.

Author Response
Dear Reviewer,
We are grateful for the opportunity to revise our manuscript. The detailed and constructive feedback from the review has been extremely beneficial in improving our work. Below, you will find our thorough response to each of their comments.
- Please insert data for main results
The data were added
- According the reviewer's suggestion of reflecting the aim of the study
- There are too many references in the introduction
We reduced part of the references
- Help your readers understand the mechanism of the main variables (Post COVID-19 vaccine, resilience and personal factors) in the introduction. Also explain why these concepts are important in Israel.
We are grateful for the reviewer this insight. We rephrased and rearranged the introduction, adding clarification and relevance for the main variables and it's relation to Israel (where relevant).
- Long descriptions and phenomena can bore readers. Delete some subheadings…
We shortened the descriptions and also removed subheadings.
- Briefly state the purpose of your research in 3-4 lines and state your hypotheses
The aim of the research was rephrased briefly and followed by the hypotheses.
- We added the title for the methods section
- Instrument – add a range of possible scores…
The minimum and maximum scores of the instruments were added to the instruments section.
- Add a description of your SES.
We added a detailed description of measuring the self-report of SES
- The paragraph "following the end of the third lockdown…".
This paragraph was moved to the introduction section according to the reviewer's suggestion.
- What was the SES status of Israel compared to the rest of the world?
The SES in our study was measured by a subjective estimation of the respondent, and relatively close to the average family income in Israel. Therefore we have no objective data regarding SES in the world.
- What intervention program are you referring to?
We refined in a general outline in the section of implications in practice our suggestion to an intervention program.
- Implications: synthesize your findings and describe the directions you can take from this research.
We rephrased and synthesized the section of implications.

Round 2
Reviewer 2 Report
Comments and Suggestions for Authors
The paper is much improved - the aurhors did their best to make the requested corrections.